# Synthesis, X-ray Characterization and Density Functional Theory (DFT) Studies of Two Polymorphs of the α,α,α,α, Isomer of Tetra-*p*-Iodophenyl Tetramethyl Calix[4]pyrrole: On the Importance of Halogen Bonds

**DOI:** 10.3390/molecules25020285

**Published:** 2020-01-10

**Authors:** Dragoș Dăbuleanu, Antonio Bauzá, Joaquín Ortega-Castro, Eduardo C. Escudero-Adán, Pablo Ballester, Antonio Frontera

**Affiliations:** 1Institute of Chemical Research of Catalonia (ICIQ), The Barcelona Institute of Science and Technology (BIST), Avinguda Països Catalans, 16, 43007 Tarragona, Spain; ddabuleanu@iciq.es (D.D.); eescudero@iciq.es (E.C.E.-A.); 2Departament de Química Analítica i Química Orgànica, Universitat Rovira i Virgili, carrer Marcel•li Domingo, 1, 43007 Tarragona, Spain; 3Departament de Química, Universitat de les Illes Balears, Crta. de Valldemossa km 7.5, 07122 Palma de Mallorca (Baleares), Spain; antonio.bauza@uib.es (A.B.); joaquin.castro@uib.es (J.O.-C.); 4Catalan Institution of Research and Advanced Studies (ICREA), Passeig Lluís Companys, 23, 08018 Barcelona, Spain

**Keywords:** polymorphs, halogen bonds, supramolecular chemistry, lattice energies, Density Functional Theory (DFT) calculations

## Abstract

This manuscript reports the improved synthesis of the α,α,α,α isomer of tetra-*p*-iodophenyl tetra-methyl calix[4]pyrrole and the X-ray characterization of two solvate polymorphs. In the solid state, the calix[4]pyrrole receptor adopts the cone conformation, including one acetonitrile molecule in its aromatic cavity by establishing four convergent hydrogen bonds between its nitrogen atom and the four pyrrole NHs of the former. The inclusion complexes pack into rods, displaying a unidirectional orientation. In turn, the rods form flat 2D-layers by alternating the orientation of their *p*-iodo substituents. The 2D layers stack on top of another, resulting in a head-to-head and tail-to-tail orientation of the complexes or their exclusive arrangement in a head-to-tail geometry. The dissimilar stacking of the layers yields two solvate polymorphs that are simultaneously present in the structures of the single crystals. The ratio of the two polymorph phases is regulated by the amount of acetonitrile added to the chloroform solutions from which the crystals grow. Halogen bonding interactions are highly relevant in the crystal lattices of the two polymorphs. We analyzed and characterized these interactions by means of density functional theory (DFT) calculations and several computational tools. Remarkably, single crystals of a solvate containing two acetonitrile molecules per calix[4]pyrrole were obtained from pure acetonitrile solution.

## 1. Introduction

Calix[4]pyrroles are well-known receptors for the binding of a variety of guests ranging from neutral Lewis bases to different anions and ion pairs [1,2,3]. In fact, calix[4]pyrroles were used as chloride transporters in liposomal models and cells [4]. Ballester’s group utilized “two-wall” α,α-aryl-extended and “four-wall” α,α,α,α-aryl-extended calix[4]pyrroles for the study and evaluation of anion-π interactions [5] and as synthetic anion transporters demonstrating that the “four-wall” calix[4]pyrroles are better carriers than the “two-wall” counterparts [6].

Polymorphism studies of calix[4]pyrrole compounds are scarce in the literature. Lynch et al. reported monoclinic (room temperature) and triclinic (low-temperature) phases of meso-octa-methylcalix[4]pyrrole complexed to dimethyl sulfoxide [7]. Moreover, Panda’s group reported two polymorphic forms of the trans isomer of meso-diacylated calix[4]pyrrole [8]. More recently, Sessler’s group reported two polymorphic forms of ion-pair receptors based on hemispherand-strapped calix[4]pyrrole derivatives [9].

Halogen bonding (XB) is currently a well-established noncovalent interaction that is similar to a hydrogen bond (HB). However, a relevant difference is the higher directionality of XB that stems from the σ-hole (small area of positive potential) of the halogen being surrounded by a belt of high electron density [10]. Therefore, a linear approximation of the electron rich atom or groups of atoms opposite to the C–X bond is required. Halogen bonding has been successfully used in supramolecular crystal engineering [11], conducting and magnetic materials [12,13] and catalysis [14].

Herein we report the improved synthesis of the tetra-α isomer of a calix[4]pyrrole bearing *p*-iodophenyl and methyl substituents in each four of its meso-carbons (see Scheme 1a). We also described the isolation of single crystals of the compound from CHCl_3_:acetonitrile solvent mixtures. The X-ray structure of the crystals revealed the simultaneous presence of two solvate polymorphic phases. Both phases crystallize in the triclinic P-1 symmetry group. However, they provide a significantly different structural arrangement of the receptor in the crystals. The importance of Type I (see Scheme 1b) halogen···halogen interactions involving the iodide atoms in the crystal packing of one of the polymorphs is studied and rationalized using density functional theory (DFT) calculations, molecular electrostatic potential (MEP) surfaces and noncovalent interaction plot index (NCIPLOT) computational tools. The single crystals that grew from pure acetonitrile solution of the receptor featured the unexpected incorporation of two acetonitrile molecules in their structure.

## 2. Results and Discussion

### 2.1. Synthesis

A few years ago, we reported the synthesis of the α,α,α,α isomer of tetra-*p*-iodopheny tetra-methyl calix[4]pyrrole in an overall yield of 16% [15]. Owing to the versatile and easy synthetic transformation of this molecular scaffold into a variety of super-aryl extended calix[4]pyrrole derivatives [16,17], we sought to optimize its preparation in a multigram scale. After numerous optimization procedures, we describe herein its synthesis in batches of more than 4 g, using high dilution conditions for the acid catalyzed condensation of the 4′-iodoacetophenone and pyrrole. We placed 500 mL of a dichloromethane (DCM) solution (0.08 M) of 4′-iodoacetophenone (10 g, 40.6 mmol) in a 1000-mL round bottom flask. Next, 5 mL of aqueous chloridric acid (36%, 40.6 mmol) were added dropwise to the above solution. With the assistance of an automatic injector pump syringe, we added a solution of 100 mL of dichloromethane containing 2.82 mL of pyrrole (40.6 mmol) to the above reaction mixture over the course of 24 h. The reaction flask was protected from light using aluminum foil and the reaction mixture was left stirring for 48 h at room temperature. A solid precipitate appeared during the reaction. The solid was filtered and washed with 300 mL of methanol. The filtered and washing organic layers were combined and concentrated under reduced pressure to afford a brown solid. The α,α,α,α isomer was obtained as a brownish solid (4.55 g, 36%) after silica column chromatography purification of the reaction crude using a 40:60 mixture of DCM:hexanes as the mobile phase.

Analytical samples of the α,α,α,α, tetra-*p*-iodophenyl tetramethyl calix[4]pyrrole were obtained by crystallization from solvent mixture containing CHCl_3_ and acetonitrile in different proportions. The single crystals that grew from the solutions were analyzed by X-ray diffraction. The solution of the diffracted data revealed the presence of the compound as two solvate polymorphic phases including only one acetonitrile molecule. Remarkably, the crystals obtained from pure acetonitrile solution displayed the incorporation of two acetonitrile molecules in the packing of the lattice.

### 2.2. Structural Description of the Packing in the Single Crystals

Figure 1 depicts the asymmetric units of the crystal structures of the solvates of polymorph **A** (left panel) and polymorph B (right panel). Both polymorphs crystalize in the triclinic P ī symmetry group. In both cases, the calix[4]pyrrole adopts the cone conformation and includes one acetonitrile molecule in its aromatic cavity through the establishment of four H-bonds with the pyrrole rings. The average CH_3_CN···N(pyrrole) distances are almost identical in both polymorphs (3.209 Å in **A** and 3.205 Å in **B**). Nevertheless, there is a subtle difference in the size of the aromatic cavities of the tetra-iodo calix[4]pyrroles in the two polymorphs. The zenithal view of the inclusion complexes (Figure 1, bottom) includes the I···I distances measured for the polymorphs. In short, the inclusion complex in polymorph **A** has two I···I distances slightly shorter (horizontal) and two slightly longer (vertical) than those in polymorph **B**. Most likely, these small geometric differences are a consequence of the dissimilar packing of the crystal lattice (vide infra).

In both polymorphs, the CH_3_CN@calix[4]pyrrole inclusion complex packs side-by-side into rods displaying an identical orientation of the *p*-iodo-substituents. In turn, the rods form 2D layers with alternating orientation of *p*-iodo substituents. Figure 2 displays side and top views of size-selected packing (3 × 3 complexes) of the 2D layers present in the crystal lattices of the two polymorphs highlighting their structural similarities.

The significant structural difference between the two polymorphs is found in the arrangement in which the 2D layers of CH_3_CN@calix[4]pyrrole inclusion complexes stack on top of another (Figure 3). In the case of polymorph **A**, the staking of the layers produces a columnar arrangement of inclusion complexes exclusively featuring a head-to-tail orientation. On the other hand, in polymorph **B**, the stack of 2D layers results in alternative head-to-head and tail-to-tail arrangement of inclusion complexes. In short, where calix[4]pyrrole units are out-of-register in polymorph **A**, producing alternating columnar stacks of unidirectional oriented molecular units. They are in register in the packing of polymorph **B**, yielding columnar stacks of dimeric capsules stabilized by four halogen-bonding interactions.

The head-to-tail or parallel orientation of inclusion complexes present in polymorph **A** leads to halogen bonds in which the acceptors unit (XB acceptor) is the electron-donor π-system of the pyrrole rings. In contrast, the *p*-iodine substituents of two CH_3_CN@calix[4]pyrrole inclusion complexes located in head-to-head (antiparallel arrangement) present in polymorph **B** are involved in Type I, C–I···I–C “like–like” halogen bonding interactions (See Scheme 1). The geometric and energetic details of both halogen-bonding interactions that are present in the two different dimeric aggregates mentioned above were further investigated using DFT calculations and the obtained results are described in detail in the next section.

Remarkably, the X-ray results showed that the two polymorphs were present simultaneously as different phases in single crystals. Moreover, the ratio of the two phases varied as a function of the content of acetonitrile in the solvent mixture used to grow crystals. Thus, at high concentrations of chloroform, the single crystals contained polymorph **A** as the major component. Conversely, as the composition of the solvent mixture increased in acetonitrile percentage, the obtained single crystals largely displayed polymorph **B**. In short, the reduction in CHCl_3_ content in the solutions used to grow the crystals favored the establishment of halogen bonding interactions between the iodine atoms of the calix[4]pyrrole units in the solid-state. The ratios of the two polymorphic phases displayed by the crystals were quantified using single crystal X-ray diffraction data. Table 1 lists the accurate obtained values.

Finally, the complete removal of CHCl_3_ from the solution used to grow crystals of the calix[4]pyrrole produced a new solvate, **C**, incorporating two acetonitrile molecules. In solvate **C**, the tetraiodo-calix[4]pyrrole receptor also adopts the cone conformation by including one hydrogen-bonded acetonitrile molecule in its aromatic cavity. Remarkably, the CH_3_CN@calix[4]pyrrole inclusion complex in the solid state of solvate **C** features two I···I distances (horizontal) that are quite similar, however the other two (vertical) are significantly dissimilar (Figure 4a). In solvates **A** and **B**, the I···I distances either in the horizontal or vertical pairs were almost identical. We assign these differences to the packing effects of the lattice. Also, in contrast to solvates **A** and **B**, displaying only the included acetonitrile molecule in the receptor’s scaffold, the asymmetric unit of solvate **C** reveals the presence of an additional molecule of acetonitrile (Figure 4a,b colored in yellow). This second acetonitrile molecule is bound in the shallow and electron-rich aromatic cavity defined by the four-pyrrole rings of the CH_3_CN@calix[4]pyrrole inclusion complex in cone conformation. This aromatic cavity possesses a suitable size for the inclusion of the methyl group of the acetonitrile molecule and establishes multiple CH-pi interactions between the methyl hydrogen atoms and the electron-rich pyrrole rings. The CH_3_CN@calix[4]pyrrole inclusion complex and its externally bound acetonitrile pack into rods having the iodo-substituents oriented in the same directions. In addition, the externally bound acetonitrile molecule is sandwiched between two twisted CH_3_CN@calix[4]pyrrole inclusion complexes of an adjacent rod. The packing of the unidirectional-oriented rods of solvated inclusion complexes form extended layers (Figure 4c). The stack of two extended layers, stabilized mainly through side-to-side C–H···π interaction of inclusion complexes, forms a dimeric layered block in which the tetra-iodo substituents of the complexes are oriented in opposite direction in order to cancel their dipoles. Finally, the dimeric-layered blocks stack on top of another also by alternating the orientation of their tetra-iodo substituents, but with a slightly shifted side-by-side arrangement of inclusion complexes. This results in the observation of stair-like 2D-layers of the CH_3_CN@calix[4]pyrrole inclusion complexes when the lattice is viewed from the b axis.

### 2.3. Theoretical Study

#### 2.3.1. Lattice Energies

First, the lattice energies for both polymorphs **A** and **B** were estimated by using a supercell of two molecules and periodic boundary conditions at the Generalized Gradient Approximation/Perdew-Burke-Ernzerhof (GGA/PBE) level of theory by means of the DMOL^3^ software. The computed values were calculated using the formula *E*_lattice_ = *E*_crystal_/n − *E*_molecule_ as recommended in the literature [18]. The resulting lattice energy values are similar for both polymorphs, that is *E*_lattice_ = −74.4 and −71.5 kcal/mol for **A** and **B**, respectively. When the relaxed coordinates are used instead of those of the X-ray crystals for the calculations, the computed lattice energies become almost identical (*E*_lattice_ = −71.9 and −71.0 kcal/mol for **A** and **B**, respectively). Experimentally polymorph **B** features the largest density.

#### 2.3.2. MEP Surface Analysis

Figure 5 shows the molecular electrostatic potential (MEP) surface computed for the calix[4]pyrrole receptor in polymorph **B**, as a model of the molecule in both of them. The MEP surface is useful to rationalize and predict donor–acceptor interactions since it identifies the electron rich and electron poor regions of the molecule. In Figure 5, we depict two zenithal views of the receptor, one with the C–I bonds pointing towards the viewer (Figure 5a) and another with the C–I bonds pointing opposite to the viewer (Figure 5b). The most positive region is located in the interior of the cavity where the four N–H bonds converge. The MEP value is very large (+69 kcal/mol) and explains the ability of this type of molecules to incorporate electron rich guests. The MEP surface also evidences that the potential energy value at the σ-holes of the I-atoms are positive and moderately strong (+16.8 kcal/mol), therefore suitable for interacting with electron-rich atoms or groups of atoms. Finally, the MEP at the surface of the aromatic H-atoms is also positive (+15.0 kcal/mol). This MEP value is similar to the one assigned to the σ–hole of I. Consequently, H-bonding interactions involving these H-atoms could compete with the formation of halogen bonds. The MEP values in the equatorial regions of the I-atoms reaches a minimum of −16.0 kcal/mol. In these equatorial region, the van der Waals surfaces of the closest I-atoms overlap. Finally, the MEP values of the π-system in the pyrrole rings are also negative (−18.8 kcal/mol). Taking into account that the interior of the cavity is unreachable by the π-system of the pyrrole ring, the most favorable interaction from a purely electrostatic point of view is the formation of halogen bonds between the I-atoms and the π-system of the pyrrole rings. This type of interaction is exclusively observed in polymorph **A** (see Figure 3a). Moreover, in the solid-state, the calix[4]pyrrole cavity already accommodates one acetonitrile molecule.

#### 2.3.3. Energetic and Noncovalent Interaction Plot (NCIPLOT) Index Analyses

Several dimeric aggregates present in the crystal lattices of polymorphs **A** and **B** were selected in order to compare their dimerization energies and correlate them with the existence of both polymorphs. Moreover, the influence of the bound acetonitrile molecule on the interaction energies was also analyzed. For polymorph **B**, we selected the two types of dimers shown in Figure 5. These dimers are responsible for the crystal growth by the packing of the 2D layers. The interactions in the growing of the 2D layers are almost identical in both polymorphs (*vide supra*) and were not analyzed in detail. In dimer 1 (Figure 6a), the I···I distances are significantly longer than the sum of the van der Waals radii (3.96 Å), thus explaining the moderate binding energy ΔE_1_ = − 9.3 kcal/mol (for six long contacts). In this type of halogen bonding (Type I), the van der Waals regions of both halogen-atoms with negligible MEP values interact (see Scheme 1a). Therefore, dispersion and polarization effects dominate this type of “like–like” halogen bonding [10]. It is interesting to note that the interaction of the dimer weakens in the absence of the bound guest acetonitrile molecules. This is likely due to the fact that the H-atoms of the methyl group of the acetonitrile interact with the negative belt of the I-atoms, thus influencing the nature and strength of the I···I interactions. The interaction energy of dimer 2 (Figure 6b) is stronger (ΔE_3_ = − 22.7 kcal/mol) because it is electrostatically more favored than dimer 1, as can be deduced from the MEP surface plot shown in Figure 4. The positive H-atoms point to the negative π-cloud of the pyrrole rings. In this case, the bound acetonitrile molecule does not affect the interaction energy to a major extend since the calculations show that the dimer stabilization only weakens 0.2 kcal/mol upon elimination of the acetonitrile molecules.

For the two dimers of polymorph **B**, the noncovalent interaction (NCI) plot index analysis has been carried out to characterize the type I C–I···I–C halogen bond and the C–H···π interactions. The NCIPLOT index is a convenient computational tool that allows for the efficient visualization and identification of noncovalent interactions [19]. Its foundation resides on the fact that the noncovalent contacts are easily identified with the peaks that emerge in the RDG (reduced density gradient) at low densities (see ref. [20] for a more comprehensive treatment). These are plotted in real space by mapping an isosurface of *s* (*s* = |∇ρ|/ρ^4/3^) for a low value of RDG. Upon formation of a supramolecular dimer, the RDG changes at the critical points in between the monomers due to the annihilation of the density gradient at these points. Therefore, the NCIPLOT index allows visualizing the extent to which NCIs stabilize a supramolecular assembly. The information that the NCIPLOT index provides is qualitative revealing which molecular regions interact. The color scheme is a red-yellow-green-blue scale with red for repulsive (ρ^+^_cut_) and blue for attractive (ρ^−^_cut_). Weak repulsive and weak attractive forces are represented by yellow and green surfaces, respectively.

The representations of the NCIPLOT index surfaces of the two dimers of polymorph **B** are shown in Figure 7. Form the two plots, it is established that the included acetonitrile molecules interact with the aromatic walls of the receptor. This is demonstrated by the presence of several green extended isosurfaces located between the acetonitrile atoms and the aromatic rings. Moreover, the interaction of the methyl H-atoms of the included acetonitrile with the I-atoms is also evidenced in the plot (exemplified in Figure 7b). More importantly, the NCIPLOT of dimer 1 confirms the existence of the six I···I interactions in spite of the long distances between the two atoms (longer than the van der Waals radii). Six symmetrically distributed globular isosurfaces are distributed between the I-atoms. The NCIPLOT index also confirms the C–H···π interactions in dimer 2, in addition to other van der Waals contacts due to the close proximity of the two molecules. The extension of the isosurfaces in this dimer suggests a strong complementarity in terms of shape, size and functionality. This result is in quite good agreement with its large computed interaction energy.

For polymorph **A**, we computed the interaction energy of a dimer extracted from its crystal lattice (Figure 8a). The interaction energy of this dimer when the acetonitrile molecules are included in the calculation is ΔE_5_ = −15.5 kcal/mol. On the other hand, removing the acetonitrile molecules reduced the interaction energy of the dimer to ΔE_6_ = −13.7 kcal/mol. This finding suggests that the existence of C–H···I interactions reinforce the halogen bonds (C–I···π) stabilizing the dimer, which are established between the host and the included acetonitrile. A likely explanation is that the electron transfer from the negative belts of the I-atoms to the acidic H-atoms of acetonitrile methyl group increases the positive MEP value at the I σ-holes, thus strengthening the halogen bond. It is also interesting to note that twice the binding energy of the dimer (head-to-tail) present in polymorph **A** (2 × ΔE_5_ = −31.0 kcal/mol) is approximately equal to the sum of the energies of the two type of dimers (head-to-head and tail-to tail) that are present in polymorph **B** (ΔE_1_ + ΔE_3_ = −32.0 kcal/mol). Keeping in mind that many other packing effects could be involved in the crystallization and final solid state architecture of the polymorphs, the similar energy values computed for the dimers detected in polymorphs **A** and **B** suggest that their crystal lattices are similarly favored, as observed by experiment. The analogous lattice energies calculated for both polymorphs also support this conclusion (see Section 2.3.1).

Finally, the NCIPLOT of the dimer of polymorph A is represented in Figure 8b that confirms the existence and relevance of the C–H···I interactions between the host and the included acetonitrile and also the C–I···π interactions involving the π-system of two pyrrole rings as donors. Both C–H···I and C–I···π interactions are characterized by green isosurfaces located between the C–I bonds and either the H-atoms of acetonitrile (C–H···I) or the π-system of pyrrole (C–I···π).

## 3. Materials and Methods

### 3.1. Materials and Techniques

The tetra-α isomer of the tetra-p-iodophenyl calix[4]pyrrole was synthesized using a modified procedure from the on reported in literature [15]. The new synthetic procedure is described in detail in the synthesis section of this paper and in the Appendix A, which also contains the spectral data of the compound.

The IR spectrum of the calix[4]pyrrole was recorded on a Bruker Optics FT-IR Alpha spectrometer (Madrid, Spain) equipped with a deuterated triglycine sulfate (DTGS) detector, KBr beamsplitter at 4 cm^−1^ resolution using a one-bounce attenuated total reflection (ATR) accessory with diamond windows. Routine ^1^H-NMR and ^13^C-NMR spectra were recorded on a Bruker Advance 400 (400 MHz for ^1^H-NMR) (Madrid, Spain) or a Bruker Advance 500 (500 MHz for ^1^H-NMR) (Madrid, Spain) ultrashield spectrometer. Deuterated solvents were purchased from Aldrich.

### 3.2. Crystalization of Polymorphs A and B

The polymorphs were present as two crystallographic phases in single crystals that grew from solutions containing different mixtures of acetonitrile (ACN) and chloroform (CHCl_3_). An aliquot of the calix[4]pyrrole isolated from the column chromatography purification of the reaction crude was dissolved in the corresponding solvent mixture at room temperature (rt). The solution was filtered and left to evaporate at rt by leaving the vial open. Single crystals obtained from the three solvent mixtures were subjected to single crystal X-ray diffraction. The solution of the diffracted data revealed the structures of the two crystallographic phases of the calix[4]pyrrole (polymorphs A and B) and the ratio in which they were present in the crystal sample. Table 1 lists the polymorphic composition of the crystals that grew from the series of solvent mixtures. The crystals that grew from pure acetonitrile solutions corresponded to a solvate of the calix[4]pyrrole containing two molecules of acetonitrile per molecule of the receptor. The packing of the lattice of this latter solvate resembles that of the polymorph solvates. However, its 2D layers show a stair-like arrangement with acetonitrile molecules intercalated between them.

### 3.3. Crystallographic Data Collection and Refinements

CrysAlisPro 1.171.40.53a (Rigaku OD, Neu-Isenburg, Germany, 2018) was used for the unit cell determination, data reduction and absorption correction. Structure solution was obtained with the program SIR2019 through the vive la difference (VLD) algorithm. Structure refinement was done with ShelXL using the ShelXLe interface. The details of the crystal parameters are summarized in Table 2. Cambridge Crystallographic Data Centre CCDC: 1971260, 1971261 and 1971262 contain the crystallographic data for **A**, **B** and **C,** respectively. Copy of the data can be obtained free of charge from CCDC, 12 Union Road, Cambridge CB2 1EZ, UK (Fax: +44-1223-336-033; E-Mail: deposit@ccdc.cam.ac.uk).

### 3.4. Computational Details

We used the X-ray geometries in the energetic characterizations of the supramolecular aggregates i.e., dimers. The level of theory used in this work was the PBE0 functional [21] in combination with the D3 Grimme’s dispersion correction [22] and the def2-TZVP basis set [23,24] by means of the Turbomole 7.2 [25] program. The molecular electrostatic potential (MEP) surfaces were obtained using Gaussian-16 [26] at the PBE1PBE/def2-TZVP level and using the 0.001 a.u. isosurface. The NCIPLOT [19,20] index has been performed using the PBE1PBE/def2-TZVP wave function.

Lattice energies (*E*_lattice_) were evaluated using the DMol^3^ software in Materials Studio 2016 [27], where all atoms were relaxed with the experimental unit cell parameters fixed. We used a double numerical with polarization (DNP) basis set as implemented in material studio [28,29]. For the solid-state calculations, PBE functional into GGA approximation [30] was utilized together with Grimme’s long-range dispersion correction [31]. Computations were carried out with the maximum number of numerical integration mesh points available and the density matrix convergence threshold being set to 10^−5^ Ha.

## 4. Conclusions

We report the improved synthesis of the α,α,α,α steroisomer of a calix[4]pyrrole framework bearing a *p*-iodophenyl and a methyl substituent in its four meso-carbons. We characterize the compound using X-ray single crystal diffraction methods and discovered the simultaneous presence of the compound as two polymorphic phases. Remarkably, the crystal lattice of polymorph **A** is dominated by conventional C–I···π halogen bonds. Conversely, polymorph **B** displays a crystal lattice dominated by type I halogen bonds and C–H···π interactions. The ratio of the two polymorphs present in single crystals depends on the solvent mixtures (ACN:CHCl_3_) used to grow them. Both polymorphs are solvates incorporating one ACN bound in the aromatic cavity of the calix. In contrast, the use of pure acetonitrile solutions produces single crystals of a new calix[4]pyrrole solvate with two ACN molecules per calix. The energetic features of the dimeric supramolecular assemblies observed in the crystal lattices of the two polymorphs are almost isoenergetic on the basis of DFT calculation results. The intermolecular interactions present in the dimers were characterized using MEP and NCIPLOT computational tools. The obtained results confirm the relevance of two different types of halogen bonds in the solid-state structure of the polymorphs. Finally, we believe that the results reported herein further support the functional relevance of halogen bonds in crystal engineering and supramolecular chemistry.

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
