# Peer review of "Synthesis, X-ray Characterization and Density Functional Theory (DFT) Studies of Two Polymorphs of the α,α,α,α, Isomer of Tetra-p-Iodophenyl Tetramethyl Calix[4]pyrrole: On the Importance of Halogen Bonds"

_molecules, 2020, doi:10.3390/molecules25020285_

Round 1

Reviewer 1 Report

The work is generally interesting. The conclusions are supported by the data and apart from minor English language editing for grammatical/spelling mistakes, the manuscript is in good shape.

For example, see Line 144 page 5 which reads " Remarkably, the X-ray results evidenced that the two polymorphs were present simultaneous as different phases in single crystals.". This is grammatically incorrect as it should read "concomitantly" or "simultaneously" instead of "simultaneous".

So I recommend that the article be accepted after minor revision and let the editorial team suggest any minor grammatical corrections such as this.

Author Response

First, we would like to thank reviewer 1 for his/her careful reading of the manuscript, and supporting publication. The changes made are listed below:

(1) We have revised the manuscript for grammatical mistakes and corrected a few of them including the one raised by the reviewer

Reviewer 2 Report

The authors reported in this manuscript about inclusion complexes between tetra-p-iodophenyl tetramethyl calix[4]pyrrole and acetonitrile. The content of the study and its explanation are rational and very interesting. Especially, the difference in packing between crystal A and crystal B is interesting. These results will get much attention of the reader of Molecules. Therefore, I recommend publication for Molecules after following minor revision.

(1) The authors must show the chemical structure of tetra-p-iodophenyl tetramethyl calix [4] pyrrole (the stereoscopic structure of it shown in supporting is preferable).

(2) A more detailed explanation of crystal C is essential.

Author Response

First, we would like to thank reviewer 2 for his/her careful reading of the manuscript, corrections and suggestions. The changes made are listed below:

(1) We have shown the chemical structure of tetra-p-iodophenyl tetramethyl calix [4] pyrrole in Scheme 1a

(2) A more detailed explanation of Crystal C has been done. That is, we have completely rewritten and expanded the text’s section describing the X-ray crystal structure of this solvate. In addition, we have also include a new Figure 4 with multiple panels in the revised version of the text. The new Figure 4 depicts in a step by step approach the different self-assembled molecular building blocks contributing to the final solid state structure observed for the bis-acetonitrile solvate.

Reviewer 3 Report

The paper by Ballester, Frontera, and co-workers reports the importance of halogen bonds in the formation of two polymorphs of calix[4]pyrrole derivatives. The paper is interesting and clearly written highlighting the importance of halogen bonds in crystal engineering and supramolecular chemistry. I think the paper can be accepted provided that a few minor details are taken into account:

- Figure 8 caption: The "(a)" part has no description. It should be mentioned that it corresponds to the optimized structure of the dimer in polymorph A (or write something similar to Figure 6).

- Figure 8b is not commented on in the main text! The authors should comment on the NCI plot, particularly on the importance of the C-I...pi bonds observed in the isosurfaces and also the existence of the C-H...I bonds in acetonitrile. Indeed the NCI plot corroborates what the authors wrote in that section.

Author Response

First, we would like to thank reviewer 3 for his/her careful reading of the manuscript, corrections and suggestions. The changes made are listed below:

(1) We have revised the caption of Figure 8 caption and described both parts (a) and (b)

(2) Thank you for taking this to our attention. Figure 8b is now commented on in the main text describing both the C-I···π bonds observed in the isosurfaces and also the existence of the C-H...I bonds involving the encapsulated acetonitrile molecule.